# Navigating the CRISPR/Cas Landscape for Enhanced Diagnosis and Treatment of Wilson’s Disease

**DOI:** 10.3390/cells13141214

**Published:** 2024-07-18

**Authors:** Woong Choi, Seongkwang Cha, Kyoungmi Kim

**Affiliations:** 1Department of Physiology, Korea University College of Medicine, Seoul 02841, Republic of Korea; choi_woong@naver.com; 2Neuroscience Research Institute, Korea University College of Medicine, Seoul 02841, Republic of Korea; 3Department of Biomedical Sciences, Korea University College of Medicine, Seoul 02841, Republic of Korea

**Keywords:** Wilson’s disease, CRISPR/Cas system, genome editing, diagnosis, gene therapy

## Abstract

The clustered regularly interspaced short palindromic repeats (CRISPR)/CRISPR-associated protein (Cas) system continues to evolve, thereby enabling more precise detection and repair of mutagenesis. The development of CRISPR/Cas-based diagnosis holds promise for high-throughput, cost-effective, and portable nucleic acid screening and genetic disease diagnosis. In addition, advancements in transportation strategies such as adeno-associated virus (AAV), lentiviral vectors, nanoparticles, and virus-like vectors (VLPs) offer synergistic insights for gene therapeutics in vivo. Wilson’s disease (WD), a copper metabolism disorder, is primarily caused by mutations in the ATPase copper transporting beta (*ATP7B*) gene. The condition is associated with the accumulation of copper in the body, leading to irreversible damage to various organs, including the liver, nervous system, kidneys, and eyes. However, the heterogeneous nature and individualized presentation of physical and neurological symptoms in WD patients pose significant challenges to accurate diagnosis. Furthermore, patients must consume copper-chelating medication throughout their lifetime. Herein, we provide a detailed description of WD and review the application of novel CRISPR-based strategies for its diagnosis and treatment, along with the challenges that need to be overcome.

## 1. Introduction

Copper is a prerequisite trace metal needed for mediating diverse physiological pathways by serving as a cofactor for various cuproenzymes such as ceruloplasmin, cytochrome c oxidase, superoxide dismutase, and many others [1,2]. Maintaining copper homeostasis is crucial, as disruptions in copper levels can lead to significant health issues. Excess copper accumulation can cause oxidative damage, liver toxicity, and neurological disorders [3,4]. The toxicity potential of copper, which contributes to oxidative damage through a Fenton-like reaction and impacts biomolecules such as low-density lipoprotein (LDL), is well known [5,6]. Excessive copper has been reported to increase the accumulation of abnormal proteins like p62 and cause stress on the endoplasmic reticulum due to an increase in unfolded proteins [7]. It has been shown that copper disrupts the TCA cycle by binding to a component of the pyruvate dehydrogenase complex, lipoylated dihydrolipoamide S-acetyltransferase (DLAT), resulting in the oligomerization of lipoylated DLAT. This mechanism, initially tested with copper ionophores, has also been found to occur with the natural copper transporter SLC31A1 (CTR1) [8]. Additionally, nuclear receptors such as FXR, RXR, HNF4α, and LRH-1 are also affected, with reduced binding to promoter response elements and decreased mRNA expression of target genes [9]. Wilson’s disease (WD) is among the copper metabolism disorders primarily caused by mutations in the ATPase copper-transporting beta (*ATP7B*) gene [10,11]. Pathogenic mutations disrupting the expression of any part of the amino acid sequence of ATP7B will curtail its activity, resulting in excessive amounts of copper in the body [12]. 

In this review, we provide an overview of the pathophysiology of WD and discuss the known genetic variations associated with the disease. We also explore the potential application of the clustered regularly interspaced short palindromic repeats (CRISPR)/CRISPR-associated (Cas) proteins (CRISPR/Cas) system, a revolutionary gene-editing technology, in the diagnosis and treatment of WD. The CRISPR/Cas system, originally a defense mechanism in prokaryotic cells against viral genomes [13], offers promising opportunities for cost-effective high-throughput nucleic acid screening. Moreover, its potential for targeted gene editing holds promise for treating inherited genetic diseases such as WD at their source [14]. The ability of the CRISPR/Cas system to identify specific nucleotide sequences may facilitate the development of novel tools that could offer high-throughput, cost-effective, and portable diagnostics involving nucleic acid screening [15]. In exploring these topics, we aim to provide deeper insights into the current understanding of WD and identify potential avenues for future research and clinical applications.

## 2. WD Caused by *ATP7B* Gene Mutations

ATP7B is a key protein in the homeostasis of copper in the body (Figure 1). It is predominantly expressed in the liver, kidney, placenta, lungs, brain, heart, muscles, pancreas, and intestines [16]. The ATP7B protein is bifunctional. It packages six copper molecules into one apoceruloplasmin protein molecule, which is then secreted into the plasma in functional ceruloplasmin (holoceruloplasmin) form and transported to various tissues as required [17]. It also aids the excretion of excess copper through interaction with chaperone protein, copper metabolism domain containing 1 (COMMD1), during the secretion of bile acid into the gastrointestinal (GI) tract via the bile canaliculus [18]. A proportion of copper taken up is stored in hepatocytes by metallothionein, a protein that is indispensable for storing and excreting copper when needed [19]. In most cells, including enterocytes, the ATP7B protein primarily regulates copper levels by acting as a buffer and sequestering copper. However, in certain cells, such as hepatocytes, ATP7B serves as the primary Cu^+^ transporter [20].

Classified as a member of the P-type ATPase family, ATP7B comprises eight transmembrane segments in the phospholipid bilayer membrane involving several domains: the phosphatase domain (A-domain TGEA motif), the phosphorylation domain (P-domain DKTGT), the ATP-binding domain (N-domain TGDN and SEPHL), and MVGDGVNDSP, which connects the ATP-binding domain to the transmembrane segment (Figure 2) [21,22,23,24]. The mechanism of copper transport by ATP7B is as follows [16,24,25]. Initially, the copper ion binds to the GG motif of the metal-binding domain (MBD) and subsequently transitions to the Cys-Pro-Cys (CPC) motif in transmembrane domain 6 (TMD6). Following this, ATP binds to the N-domain, thereby triggering ATP hydrolysis and phosphorylation of the P-domain. Simultaneously, ADP is released and forms an acyl-phosphate intermediate. The copper ion is then transported across the membrane by utilizing ATP as an energy source. Finally, the copper ion translocates across the membrane and the P-domain is dephosphorylated by the A-domain.

The ATP7B protein is encoded by the *ATP7B* gene, comprising 21 exons and 20 introns situated on chromosome 13q14.3 [16]. Pathogenic mutations in any part of the nucleotide sequence of *ATP7B* (Figure 3) will disrupt the synthesis and functionality of ATP7B [26,27]. The distribution of mutation types is predominantly concentrated on missense mutations (48%), followed by frameshift (24%), intronic (splicing) (10%), nonsense (9%), deletion (7%), large deletion, and promotor mutations (1% each) [28,29]. Many researchers have tried to uncover the predominant mutation variant of a specific geographical region or comprehensively analyze large amounts of data from previous studies conducted in various countries [30,31,32] (Table 1). The p.H1069Q mutation variant is predominant in most European countries, whereas the p.R778L mutation variant is prevalent in most Asian countries. However, there are still some variations in the mutation or the allele frequency in the variants of WD between countries even when they are on the same continent or between specific regions within a country. 

## 3. Utilizing the CRISPR/Cas System for the Diagnosis of WD

In recent years, advances in exploiting the CRISPR/Cas system have made it a highly precise method for the diagnosis of genetic diseases. Multiple Cas subtypes exist, such as function-spanning helicase, exonuclease, endonuclease, and endoribonuclease, among others, some of which still need to be validated [85,86]. The CRISPR/Cas system works via the collaboration of single-guide RNA (sgRNA) and a Cas protein [87]. Following the guidance of sgRNA comprising CRISPR RNA (crRNA) and trans-activating RNA (tracrRNA), the Cas protein recognizes a specific region in the genome and induces double-strand breaks (DSBs) in the target sequence on the DNA. The CRISPR/Cas system is subclassified into Class 1, which needs a multi-subunit crRNA effector complex to function, and Class 2, which needs a single crRNA effector complex to function (Table 2). Based on the difference between the specific mechanisms in the CRISPR/Cas-mediated immune response, they diverge into Types I, III, and IV (Class 1) or Types II, V, and VI (Class 2) [88]. The target substrate, protospacer adjacent motif (PAM) sequence, and cleavage cut structure vary across the CRISPR/Cas system types, thereby enabling broader application of the system [89,90]. Figure 4 shows the characteristics of the individual CRISPR/Cas system variants [89,90,91,92,93,94,95,96,97,98,99,100,101,102,103,104].

The CRISPR/Cas system is emerging as a useful diagnostic tool for genetic mutation diseases in which most mutation variants are point mutations such as WD. Associated techniques for SNP detection or that enable genomic DNA (gDNA) genotyping, including those that have been applied in other than human cell lines or tissue, have been exploited. The Class II CRISPR/Cas system, including Types II, V, and VI, is a valuable diagnostic tool [105]. The diagnostic methods employed exploit the distinctive characteristics of the various CRISPR/Cas systems, which can be categorized according to their activity mechanism into three main groups: sequence-specific binding affinity, sequence-specific DNA cleavage, and trans-cleavage activity [106].

Type II CRISPR-induced diagnostics use Cas9 or nickase Cas9 (nCas9). Cas9 introduces DSBs, while nCas9 introduces a nick in a single strand of dsDNA. This nick is achieved via a mutation designed to inactivate the catalytic domain of the Cas9 protein (D10A or H840A mutation) regarding the HNH or RuvC domains, respectively [105]. Alternatively, dead Cas9 (dCas9) can be utilized, which is the Cas9 protein with mutations in both the HNH and RuvC domains [107]. The first diagnostic strategy utilizes the ability of the Cas9 protein to cleave the target sequence. One diagnostic method employs SNP-induced CRISPR/Cas cleavage activation by introducing a novel PAM sequence [108,109]. The Cas9 protein can recognize this newly introduced PAM sequence, which arises because of the mutation, and utilizes it to facilitate the cleavage of the target gene. Conversely, the absence of the PAM sequence or the mismatch of the seed region on crRNA induced due to the mutation can also be employed for diagnostic purposes. For SNP detection, common examples of this are the surveyor assay [110], the T7 endonuclease 1 (T7E1) mismatch detection assay [111], and the polyacrylamide gel electrophoresis-based (PAGE)-based genotyping assay [112]. Recently, engineered CRISPR/Cas-derived RNA-guided nucleases with the restriction fragment length polymorphism (RGEN-RFLP) mechanism have been introduced to identify point mutations or insertions and deletions (indels) [113,114]. The RGEN-RFLP method enables the detection of homozygous mutant clones that contain identical biallelic indel sequences. Moreover, it is not limited by sequence polymorphisms near the nuclease target sites, which is achieved by inhibiting CRISPR/Cas cleavage induced by sgRNA mismatch. In light of the successful genetic analysis of oncogenic point mutations in the *KRAS*, *PIK3CA*, and *IDH1* genes of human cancer cell lines using RGEN-RFLP [114], it can be inferred that WD diagnosis should be possible by using this method.

Either nCas9 or dCas9 are used in the second diagnostic strategy involving Type II CRISPR. The CRISPR-Chip is a biosensor that uses the gene-targeting capacity of dCas9 (which can specifically bind to DNA without cleaving it) combined with sgRNA fixed on a transistor. This setup produces electric signals when the desired DNA sequence is detected [115]. The improved efficiency of the CRISPR-SNP-Chip, an advancement of the CRISPR-Chip that recruits different Cas9 orthologues and has more types of electrical measurements, has been demonstrated on DNA samples from sickle cell disease patients [116]. Other diagnostic strategies include the CRISPR-Cas9-triggered strand displacement amplification (CRISDA) method [117], finding low abundance sequences by hybridization (FLASH) [118], CRISPR/Cas9 triggered isothermal exponential amplification reaction (CAS-EXPAR) [119], and Cas9 nickase-based amplification reaction [120]. These involve amplifying amplicons including the target sequence, thereby leading to gDNA genotyping or SNP detection with considerable sensitivity and specificity. Other than DNA genotyping or SNP detection, CRISPR, sometimes combined with NGS, has also been used for nucleic acid quantification and sequencing. Methods such as nanopore Cas9-targeted sequencing (nCATS) [121]; short tandem repeat identification, quantification, and evaluation (STRique) [122]; and the depletion of abundant sequences by hybridization (DASH) [123] have demonstrated better amplification of substrates for sequencing compared to classical PCR [124]. For the identification of specific sequences, DNA fluorescence in situ hybridization (FISH) can be used in conjunction with green fluorescent protein-tagged dCas9 protein with sgRNA [125].

Type V CRISPR/Cas diagnostics, which comprise Cas12a, Cas12b, or Cas14 (Cas12f) are used, include tools named DETECTR [126], Cas14-DETECTR [127], HOLMES [128], HOLMESv2 [129], CDetection [130], E-CRISPR [131], and CRISPR-responsive hydrogels [132,133]. Although the details of their precise mechanisms differ in detail, exploitation of the trans-cleavage activity of Cas proteins other than Cas9 is used in each case. As they specifically bind with matched dsDNA substrates, their trans-cleavage activity cleaves ssDNAs nonspecifically (specificity varies by subtype); the DNA is usually provided in annealed form with fluorescent probes or in a structure that is detectable when trans-cleavage happens [134]. Type VI systems target RNAs by exploiting Cas13a (also known as C2c2), which is unique in that it possesses dual RNase functions that can mediate both RNA-guided RNA degradation and CRISPR RNA maturation [135,136]. Its nonspecific trans-cleavage activity, which works on collateral ssRNAs, is utilized in the Specific High Sensitivity Enzymatic Reporter (SHERLOCK) system [137]. Through target RNA sequence amplification by applying recombinase polymerase amplification (RPA) to dsDNA followed by transcription, Cas13a-mediated DNA detection is enabled via similar trans-cleavage involving the mechanisms used in the type V CRISPR/Cas system [138]. This was later developed into SHERLOCK v2 [139], thereby validating its potential for SNP diagnosis. 

The differences between the CRISPR/Cas systems for identifying specific mutations in the *ATP7B* gene highlight the importance of selecting the appropriate system based on the *ATP7B* variant (Figure 5A). According to the intention of the user, the Cas protein can be selected for each variant. Variants can potentially be identified by assessing whether cleavage of the mutation locus occurs within the recognized seed regions of the sgRNAs. Alternatively, variants could be identified by the presence of fluorescent activity indicative of trans-cleavage activity (Figure 5B). For the c.2333G>T (p.R778L) mutation, the Cas protein cannot recognize the target sequence due to the apparent mismatch in the seed region of the sgRNA. In the c.3207C>A (p.H1069Q) mutation, the loss of the PAM sequence due to mutation prevents the Cas protein from cleaving the target sequence on the *ATP7B* gene. Conversely, in the case of the c.865C>T (p.Q289*) mutation, the mutation generates a new PAM sequence, thereby enabling the Cas protein to cut the reporter oligonucleotide via trans-cleavage activity. For the c.2621C>T (p.A874V) mutation, the Cas protein is unable to cleave the reporter oligonucleotides because the mutation in the seed region hinders the recognition of the sgRNA.

Given that the cDNA sequence of the *ATP7B* gene contains 1,014 5′-NGG sequences, it is possible that most mutation variants can be identified by simply altering the sgRNA while using a single Cas protein, SpCas9. For example, the major variants c.2333G>T (p.R778L) and c.3207C>A (p.H1069Q) have several 5′-NGG sequences near the mutation point, respectively (Figure 5B). However, certain mutation variants of the *ATP7B* gene lack a 5′-NGG site, or the mutation point is not located within the sgRNA seed region. For instance, if a WD patient has both c.2072G>T (p.G691V) and c.2108G>A (p.C703Y) mutations, the c.2072G>T (p.G691V) mutation is likely to be recognized by SpCas9, while the c.2108G>A (p.C703Y) mutation is not. In such cases, different Cas proteins would need to be utilized (Figure 5C). For example, type V-B AacCas12b, derived from *Alicyclobacillus acidoterrestris*, which requires a 5′-TTN PAM in upstream of the target sequence, could be considered as an alternative [140].

There may be multiple PAM sites near the location of the mutation, allowing us to select different candidates for mutation detection depending on the PAM sequence location. The figure shows an example of two sgRNA candidates for diagnosing the c.2333G>T (p.R778L) variation using the RGEN-RFLP method (Figure 5D). When selecting sgRNA candidate 1 for the wild type (WT), SpCas9 cleaves the WT DNA, whereas the WD DNA remains without cleavage. In contrast, when sgRNA candidate 2 is selected for the WT, the PAM sequence is preserved in both WT and WD; however, a mismatch in the seed region results in the same cleavage pattern observed with candidate 1. Besides the two candidates shown in Figure 5D, an alternative sgRNA sequence can be selected to distinguish the mutant variant. While the theoretical predictions presented in this paper are well founded, they may yield unexpected outcomes in experimental settings. These could include off-target effects, varying cleavage efficiencies depending on the sgRNA candidates, and other factors. Therefore, rigorous experimental studies are required to identify the optimal protocol that ensures the most reliable diagnosis using the various available CRISPR/Cas system options.

## 4. Comparison of the Diagnostic Approaches

Accurate diagnosis of WD is paramount for effective treatment and management of the disease. The clinical presentation of WD typically occurs at major sites such as the liver (42%) and brain (34%) [141], and sometimes at minor sites such as the eyes, kidneys, heart, endocrine system, and musculoskeletal system. These minor sites should not be neglected, as they may occasionally present important features [142,143,144,145]. The heterogeneous nature and individual presentation of physical and neurological symptoms in WD patients pose a major challenge to accurate diagnosis. Current diagnostic approaches rely on a combination of clinical manifestation and laboratory testing [146,147]. Physicians consider clinical symptoms such as unexplained liver disease, neurological disorders, psychiatric symptoms, and Kayser–Fleischer (K-F) rings, which appear to encircle the corneas in the eyes. Laboratory tests for biomarkers such as serum ceruloplasmin (CPN), the aspartate aminotransferase/alanine aminotransferase (AST/ALT) ratio, and the alkaline phosphatase (ALP)/total bilirubin ratio, along with MRI imaging and 24-h urine copper measurements, are commonly employed. A relative exchange copper (REC) has been proposed as an alternative with superior sensitivity and specificity, determined by the ratio of exchangeable copper (CuEXC) to total copper in patient serum [148,149,150]. While it does result in increased costs compared to conventional biochemical tests due to the CuEXC determination process, the cost of the REC assay is still much more affordable than genetic analysis [146]. However, it should be noted that these biochemical tests do not provide information regarding aberrant *ATP7B*, which may be the fundamental cause of the observed symptoms.

Genetic analysis is recommended to confirm WD if diagnosis through clinical representations or biochemical tests alone is insufficient. This typically involves genetic analysis of the *ATP7B* gene using various techniques such as Sanger sequencing and next-generation sequencing (NGS) [151]. However, due to the necessity of standard molecular biology workflows, limitations in amplicon length (less than 1 kbp), the time-consuming and expensive equipment and reagents, and the accompanying bioinformatics analysis, these methods are less accessible compared to biochemical testing [146,152]. Furthermore, NGS needs highly trained personnel and special equipment that may not be affordable for some countries. In contrast, the CRISPR system has the potential to be easily applied with simple laboratory equipment, and it could be a cost-effective solution because it can recognize various target sequences by changing the sgRNA sequence as needed. Therefore, CRISPR-based diagnostics are expected to provide cost-effectiveness by alleviating health inequities and bypassing the need for complete genomic sequencing in low-resource countries, as demonstrated in the case of SARS-CoV-2 diagnosis [153,154]. Another advantage of CRISPR is its relatively short analysis time with cost reduction. RGEN-RFLP analysis can be used for genotyping without special equipment and takes approximately 90 min [114]. The experimental times for both SHERLOCK and DETECTR are relatively brief, with durations of less than two hours [137,155]. Therefore, the CRISPR-based diagnostic methods could potentially assist in the early diagnosis of WD by facilitating the identification of mutations in *ATP7B* in the medical workplace.

The CRISPR/Cas system typically achieves specificity through the one-to-one correspondence between the gRNA and the target sequence. However, when targeting hundreds of mutational variants, as may be the case with WD, the need to design hundreds of gRNAs hinders the feasibility of CRISPR diagnostics. Targeting only the predominant variants restricts the utility of CRISPR-based diagnostics to a limited number of regions. Although the development of a single CRISPR/Cas-based assay that enables simple and quick genotyping of WD pathogenic mutations has not yet been achieved, several approaches show promise in addressing this challenge. One approach involves the efficient generation of multi-targeting gRNA sets, while another employs diverse Cas protein variants or sgRNAs to target multiple sequences in a single assay.

The CRISPR-broad system attempts to overcome this limitation with an algorithm that identifies multi-targetable genomic regions and designs multi-target gRNAs, enabling the observation of broad genome regions using fluorescent proteins attached to dead Cas9 proteins [156]. Another study, the MINORg system, was aimed at generating gRNA sets for multiple homologous sequences. It succeeded in knocking out multiple homologous genes by analyzing them with variable thresholds of mismatch tolerance of gRNAs [157]. Multi-targeting sets of sgRNAs may be assisted by studies concerning the delivery of multiple sgRNAs simultaneously, which has shown proper working for each gRNA [158]. Additionally, many approaches towards efficient transportation, including Cpf1-based crRNA array [159], a multiple sgRNA cassette incorporated vector [160], or a single artificial structure that is later processed into diverse sgRNAs, are being made, which can be better than simple co-transportation of individual sgRNAs or sgRNA encoding vectors [161,162]. The multiplex application of the CRISPR/Cas system has the potential to broaden the scope of CRISPR diagnostics. 

SHERLOCKv2 leverages diverse Cas enzymes combined with different reporter molecules, resulting in varying fluorophores to identify multiple sequences [139]. Similarly, other studies have demonstrated the potential of human DNA genotyping using varying fluorophores with a CRISPR/Cas12a biochip. This biochip, combined with different sgRNAs, reporter molecules, and an inserted PAM sequence, expands its range of action over sequences without a PAM in the appropriate location [163]. Additionally, the Combinatorial Arrayed Reactions for Multiplexed Evaluation of Nucleic Acids (CARMEN) tool, combined with Cas13, utilizes assigned color codes for the detection of multiple pathogens [164]. The CRISPR/Cas system has also been employed for in vitro protein interaction studies, such as PICASSO, which uses a microarray surface combined with DNA sequences complementary to each peptide’s sgRNA, enabling the analysis of multiple peptides in a single assay [165]. The generation of multi-targeting gRNA sets or the analysis of complex signals generated by multiple reporters may both assist in identifying target variants hidden among hundreds of mutations.

## 5. CRISPR/Cas System-Based Gene-Editing Therapy Strategies for WD

Currently used treatments are mainly focused on decreasing copper absorption and increasing copper excretion. As copper is absorbed through the diet, it is advised that patients switch to a low-copper diet. However, copper restriction is now regarded less strictly than in the past due to insufficient evidence [166,167]. Recently, more emphasis has been placed on medical treatment, complemented by moderate dietary changes [168]. The usual daily limit for copper ingestion is 1 mg, which can be adjusted according to each patient’s condition after medical consultation. Low-copper foods (<0.08 mg per serving) and medium-copper foods (0.08–2.0 mg per serving) can be eaten in adequate amounts for nutrition balance. High-copper foods (copper > 0.2 mg per serving), including vegetable juices, sweet potatoes, dried fruits, soy flour, soy milk, dried beans, and desserts comprising candy with nuts and/or cocoa, among many others, are not recommended. The most frequently employed medical treatments include D-penicillamine, trientine, zinc, sodium dimercaptopropane sulfonate, and dimercaptosuccinic acid (Table 3).

While dietary control and chelation of copper have shown promising results, it is important to consider the long-term implications of these treatments. Patients must adhere to a low-copper diet or take copper-chelating medicine for the rest of their lives, which may not be feasible in some cases. It coerces them to be continuously concerned about possible adverse effects that may come from copper chelation drugs. While dietary control is recommended for all WD patients, it will be insufficient to hinder the progression of the disease if the patient depends on it alone without medical treatment [167,192]. Therefore, it is worth exploring the CRISPR/Cas system for treating WD to enhance patients’ quality of life.

The Cas protein derived from *Streptococcus pyogenes* (SpCas9), which is one of the most widely used in the CRISPR/Cas system for gene editing, has distinct nuclease domains (HNH or RuvC) for cleaving each strand of dsDNA [193]. It forms a complex with the target crRNA annealed to tracrRNA needed either for the maturation of pre-crRNA or the action of the SpCas9 complex; this recognizes the complementary target sequence or the protospacer that is complementary to the spacer of the crRNA [194,195]. SpCas9 makes a DSB in the dsDNA 3 bp upstream from PAM sequence 5′-NGG. The RuvC domain works at the non-target DNA strand while HNH works at the target strand [100]. The DSB is repaired by endogenous DNA repair mechanisms, nonhomologous end joining (NHEJ), or the homology-directed repair (HDR) pathway. As mentioned earlier, DSB repair is carried out by either NHEJ, which introduces mutations as small indels, or HDR enabling precise base editing for the controlled introduction of mutations using DNA donors [196,197]. The error-prone NHEJ pathway is independent of the cell cycle and results in frequent indel mutations that may often cause loss of function, while high-fidelity HDR can only be used in the G2 or S phase for precise gene modification [193] (Figure 6A). Several researchers have shown the possibility of *ATP7B* gene editing with Cas9 via HDR [198,199,200,201]. For instance, Miyaoka et al. [200] showed that the p.P778L (c.2333G>T) mutation in the wild-type *ATP7B* gene was induced in the HEK293T cell line using Cas9 with single-strand oligodeoxynucleotide (ssODN)-donor DNA. In another study, the potential for restoring copper exportation capability has been demonstrated in a WD mouse model through the transplantation of gene-edited pluripotent stem cell-derived hepatocytes (iHEPS) co-transfected with an ssODN and a plasmid containing sgRNA and Cas9 via electroporation [202]. In addition, the transfer of CRISPR/Cas9 with ssODN by lipofection was also successful in restoring copper transportation and copper resistance by up to 60% in human cell line HEK293T [203]. However, because of the potential detrimental effects that DSB can cause, such as undesired edits by NHEJ, the application of base editing without generation of DSBs has been studied, utilizing base editors (BEs) and prime editors (PEs) with improved specificity and efficiency [204]. 

Recently described BE and PE techniques may provide therapeutic strategies for the precise, efficient, permanent, and safe correction of pathogenic mutations in humans [205]. BEs and PEs are made by partially or completely inactivating the cleavage domains of Cas proteins (nCas or dCas) in which the cleavage activity is limited to a unilateral strand or completely diminished while well preserving binding specificity [107,206,207,208]. It applies not only to the DSB-inducing Type II and V CRISPR/Cas systems but also to Type VI, which targets ssRNA [209].

BEs are primarily categorized as adenine base editors (ABEs) and cytosine base editors (CBEs) (Figure 6B and Figure 6C, respectively). CBEs facilitate the conversion of a C/G base pair to a T/A base pair, and vice versa for ABEs [206,207]. A CBE is made by engineering gRNA-nCas9 ribonucleoprotein (RNP) to become complexed with cytidine deaminase and uracil glycosylase inhibitor (UGI) [207]. Cytidine deaminase converts cytosine to uracil, recognized as thymine in the cellular mismatch-repair pathway [210,211]. UGI is needed to inhibit uracil DNA glycosylase (UDG)-induced uracil removal. To direct the mismatch-repair mechanism toward the resynthesis of the desired non-target strand instead of the target strand, nCas9 makes a nick on the non-target strand, resulting in a U/A base pair that is later converted to the desired T/A base pair [212]. The working mechanism for ABE is similar to that for CBE, except that cytidine deaminase (a member of the APOBEC (apolipoprotein B mRNA-editing enzyme, catalytic polypeptide) family) is replaced with adenine deaminase made to artificially work on DNA substrates by engineering *Escherichia coli*-originating TadA (ecTadA), a tRNA adenine deaminase, that converts adenine to inosine, which is read as G by polymerases [206,213]. Different choices of Cas protein and improvement of deaminases vary the properties of BE, so appropriate selection depending on the application is needed. The BE class (CBE or ABE), editing window, editing efficiency based on the sequencing context, off-target editing, and Cas PAM existence are some of the factors that need to be considered [214]. Recently, improved BEs that expand on the available base substitutions by CBE or ABE have also appeared, such as a C-to-G BE (CGBE) [215], an A-to-Y BE (AYBE) [216], and other diverse types [216,217,218]. Liu et al. [219] attempted to generate and correct pathogenic mutations c.3097G>A (p.T1033A) and c.3659C>T (p.T1220M) in the *ATP7B* gene by using xCas9-derived BEs that recognize the NGH PAM sequence in HEK293T cells. They were able to induce T1033A and T1220M via A-to-G substitution using BPNLS-xABE and C-to-T substitution using BPNLS-Gam-xBE3, respectively. They then rescued each mutation using different sgRNAs to correct the generated mutations. The editing efficiencies were 20% and 40% for T1033A and T1220M, respectively.

While the introduction of BEs has marked a significant advancement in precise gene editing, several limitations persist. These include bystander editing resulting from narrow editing windows, restricted availability for only synonymous editing, off-target editing, and the inability to address certain pathogenic mutations involving small or large indels [220,221,222,223]. This has led to the development of novel gene-editing tools called PEs, which allow for precise correction of all types of mutations without generating DSBs [224] (Figure 6D). A PE is a complex comprising nCas9, reverse transcriptase (RT), and a prime-editing guide RNA (pegRNA). The prime-editing process involves nicking the non-target strand, which generates 5′ and 3′ flaps on the ssDNA strand. The 3′ flap is then elongated via reverse transcription using the extended pegRNA as an RNA template. Subsequent hybridization and stabilization of the dsDNA occur after cleavage of the 5′ flap, followed by mismatch repair in the desired direction, resulting in edit-incorporated dsDNA [225]. Prime editing has the advantage of enabling not only the substitution of a single base pair but also the inducement of a small indel in the target gene using the editing sequence in the pegRNA. For example, prime editing was used to correct a single base pair duplication (c.1288up, p.S430fs) in the *ATP7B* gene by inducing a single base pair deletion in an organoid system [226]. However, the applicability of prime editing is being hampered by low editing efficiency that varies according to the target locus and cell type. Proposed improvements to overcome these limitations include relaxed PAM restriction [227,228,229,230,231], enhanced editing efficiency [232,233,234,235], and protection from early degradation [236]. Table 4 shows the advantages and limitations of the currently attempted CRISPR-based approaches for WD.

While we have largely discussed the promising future anticipated from the development of CRISPR-Cas technology, there are also conflicting views. The most concerning and frequently mentioned adverse effect is off-target editing. Undesired mutagenesis originating from off-target or bystander edits could induce genetic hazards [237,238], including not only small indels at on-target or off-target loci but also large deletions with chromosomal translocations [239,240,241]. Efforts to avoid undesired mutations are ongoing, and engineering CRISPR/Cas tools from the initial stage has enhanced safety [242,243,244,245]. In parallel with efforts to reduce off-target and bystander edits, diverse approaches for off-target detection and prediction have been developed [246,247]. 

On-target cytotoxicity, arising from faulty repair of generated DSBs, indicates the potential for chromosomal structural variants (SVs) [248,249,250]. The generation of chromosomal SVs has already been reported in human cell lines, including cancer cell lines, primary cells, and embryonic cells [251]. Several studies have attempted to engineer the Cas protein to enhance the frequency of HDR while inhibiting NHEJ [252,253,254,255]. Tools like BE and PE offer another precise method of gene editing without DSBs [256,257]. Nevertheless, further investigation is necessary to eliminate undesired editing.

Another limitation of CRISPR/Cas is the mandatory need for PAM sequences and the existence of narrow editing windows [206,231,258]. Strategies to enhance editing efficiency include engineering the Cas protein and components involved in related cellular mechanisms, designing additional sgRNAs, and activating the transcriptional activation domain [259,260,261,262]. Although recent trials of gene-editing treatments for WD have shown promising results, further research and refinement are necessary to address the remaining challenges before their clinical application can be fully accepted.

The high cost of the medicine is also one of the limitations of gene therapy. While many CRISPR diagnostics offer the promise of low cost, CRISPR-based gene therapy does not. Multiple factors contribute to this problem, including the disadvantages faced by minority populations, limited access to cutting-edge therapies, and high costs. For example, the successful CRISPR-Cas9 targeted gene editing of CD34+ hematopoietic stem and progenitor cells has shown potential as a treatment for transfusion-dependent β-thalassemia and sickle cell disease (SCD) [263]. However, it does not necessarily ensure an advantage for those most affected by SCD due to limited research and funding for this condition [264,265]. The novel CRISPR-based gene therapy for SCD, exa-cel, is predicted to cost $4–6 million for lifetime treatment, far exceeding the costs of most previously approved gene therapies [266]. Therefore, it is becoming increasingly clear that many patients will face financial challenges in paying for healthcare services. Considering this, innovative insurance models have been proposed, including medical mortgages, refunds for suboptimal results, and subscriptions [267].

The application of CRISPR/Cas gene editing on human embryos and the reproductive system is deeply intertwined with long-term safety and ethical issues, given the already existing state of genomic instability [268,269,270]. Proponents of embryo-targeted genome editing technologies argue that gentle modification of target genes is now possible, claiming that lifelong medication and other supports might be more challenging to afford than gene therapy, which eliminates the genetic mutation from the embryo and future descendants [271,272]. As the dispute surrounding the clinical application of CRISPR/Cas gene therapy involves several complex issues that are unlikely to be easily resolved, it is crucial to provide accurate information on both the pros and cons to patients who wish to undergo such treatments.

## 6. The Current Stage of Vector Development for Gene Therapy to Treat WD

Although CRISPR/Cas-induced DSB, BEs, and PEs are currently the most widely used forms of gene therapy, the accurate delivery of the gene-editing tool to the target cells remains a significant challenge. Electroporation [202,226] and lipofection [198,199,200,203,219] are well-developed delivery systems for gene transportation into living cells. Nevertheless, although they have both demonstrated satisfactory outcomes in the gene editing of the *ATP7B* gene in vitro, their direct application in vivo is challenging, which presents a significant obstacle to their clinical implementation. To date, many CRISPR/Cas system-based gene therapies have been investigated using in vitro electroporation and lipofection, while most in vivo experiments have been conducted using the transfection of cDNA for the *ATP7B* gene (Table 5). Thus, developing an effective delivery system remains a crucial pillar for advancing CRISPR/Cas system-based gene-editing technology.

Viral vectors have emerged as an alternative approach to address the limitations associated with direct transfection in living cells in vivo. One of the most actively researched vectors is the adeno-associated virus (AAV) vector. AAV, a member of the *Dependoparvovirus* genus, can carry approximately 4.8 kb of ssDNA and is considered to be relatively safe due to its reliance on co-infection with adenovirus for replication, which triggers a mild immune response [273]. Its viral vector comprises engineered recombinant AAV (rAAV), in which the viral DNA is replaced with the DNA cargo for intracellular delivery. In rAAV, the *Rep* gene is typically omitted and the DNA cargo is flanked by inverted terminal repeats (ITR), thereby forming circular double-stranded episomal DNA [274]; the latter avoids integration into the host cell’s genome, which could lead to undesired mutations. In addition, various AAV serotypes (AAV1 to AAV9) exhibit distinct tissue tropisms and receptor specificities, thus enabling the efficient targeting of the desired tissue [275]. However, the limitations of AAV technology include the capacity of the AAV vector, (approximately 4.8 kb, albeit this differs slightly according to the AAV serotype): the larger the DNA cargo, the greater the retardation of vector formation [276]. Moreover, as it takes some time to synthesize the AAV vector into a double-stranded episome after entering the cell nucleus, it is not suitable for inducing rapid outcomes [277]. Although self-complementary AAV (scAAV) can achieve faster onsets [278], the packaging capacity is cut by more than half (~2.2 kb) [279]. The transient effect of AAV that degrades following the repopulation of tissue cells offers both pros and cons in the therapeutic approach [280]. Several strategies have been explored to overcome these limitations. For instance, efforts have been made to enhance productivity by reducing the dimensions of the AAV vector by introducing a mini-*ATP7B* gene, the expression of which results in MBDs 1–4 being truncated [281,282,283]. Since these MDBs contribute less to the activity of ATP7B than the others, the result is effective liver function restoration in WD model mice, albeit inferior to fully functional wild-type ATP7B. In another study [284], split-intein technology was used to facilitate the intracellular delivery of the full *ATP7B* gene to the liver of mice. Furthermore, it has been reported that significant improvements in a mouse WD model were achieved by loading a codon-optimized *ATP7B* gene onto AAV8 vectors with reduced-sized transthyretin enhancer and promoter sequences to enhance the expression of *ATP7B* [285]. Recently, clinical trials have been conducted on two novel AAV-related gene therapies: UX701, which is an AAV9 gene therapy used in a seamless phase 1/2/3 study design for Cyprus2+ [286,287], and VTX-801, which is a gene therapy utilizing nonreplicating rAAV with a mini-*ATP7B* gene [286]. 

Lentivirus is an ssRNA retrovirus that replicates through the *gag*, *pol*, and *env* genes which encode structural proteins, RT and integrase, and the viral envelop glycoprotein, respectively. Following the fusion of the viral envelope with the cell membrane, the viral core is released into the cytoplasm, where it undergoes reverse transcription to form dsDNA and subsequently translocates to the nucleus. Integration of the viral dsDNA with the host cell’s genome occurs, thereby forming a provirus that unfortunately poses the risk of introducing undesired genetic mutations. The transcribed viral RNA is then shuttled back to the cytoplasm, where it undergoes translation, leading to it being packaged into viral particles. These particles are then exported extracellularly after undergoing final maturation [288]. Several modifications have been implemented to utilize lentivirus as a gene-delivery vehicle. The first-generation lentiviral vector was engineered to contain a segment of the HIV genome, the *gag* and *pol* genes, and additional viral proteins enveloped by VSV-G for tropism against diverse types of cells. Lentiviral accessory genes *vif*, *vpr*, *vpu*, and *nef*, which enhance survival, were included but then excluded in second-generation lentiviral vectors without hindering the transfer process. In addition, the *tat* and *rev* genes essential for replication were included in the first-generation vectors. The incorporation of viral functions into the budding particles was markedly reduced as a result of a mutation in the 5′-untranslated region of the retroviral genome [289]. In third-generation vectors, the viral genome was split into separate segments, thereby enhancing the safety of the vector. In addition, long-terminal repeats (LTRs), which are indispensable for the integration of vectors, were mutated to self-inactivating (SIN) lentiviral vectors for the prevention of insertional mutagenesis [290]. In Merle et al.’s study [291], the authors used a lentiviral vector to transduce hepatocytes with full *ATP7B* cDNA and transplanted them ex vivo into the lymphatic endothelial cells (LECs) of mice. Hepatic expression of the transgene remained up to the end of the experiment (24 weeks), validating the conclusion that long-lasting gene expression is possible with appropriate phenotypic improvement, which could be an advantage over AAV vectors despite the risk of insertional mutagenesis. This could provide a novel approach to the treatment of WD.

Although virus-like particles (VLPs) can self-assemble into a structure that resembles the original virus by expressing molecules constituting the viral capsid, core, or envelope, they are incapable of replication as they lack the viral genome [292]. They have been divided into enveloped and non-enveloped VLPs and then subdivided into the number of capsid layers and the number of types of proteins that constitute the capsid [293,294]. Generally, VLPs have been employed to counter infectious pathogens and in cancer vaccination [295]. However, they can also be utilized as drug or nucleic acid carriers under certain circumstances, for which they have exhibited superior performance over other nanoparticles because of their marked resistance to lysosomal degradation [296]. Moreover, a specific on-target effect can be realized by engineering the surface receptors or utilizing tropism resembling that of the original virus. Although an investigation of the specific application of VLPs loaded with the CRISPR/Cas system on a WD model has not yet been conducted, the advancement of VLPs and their application to hepatic tissue has yielded promising results. These findings have been validated through in vitro gene transfer experiments using HEV-derived VLPs on liver-derived cell lines [297]. Murine leukemia virus-like particles incorporated with Cas9-sgRNA RNPs (nanoblades) have also demonstrated efficient in vivo editing of the *Hpd* gene in mouse liver. These nanoblades have been further utilized in complexes with donor DNA for HDR or employing modified Cas9 variants for transcriptional upregulation [298]. A recently developed version of BE-engineered VLP (BE-eVLP) was able to introduce nuclear export signals (NESs) and optimize the *gag*-BE: *gag-pro-pol* stoichiometry; it achieved 60% A-to-G conversion in the *Pcsk9* gene in mouse liver [299]. While prior studies have not been focused on WD models, this approach should be applicable.

Apart from the biological transportation vectors explained above, chemical methods such as lipid nanoparticles (LNPs) have also been used for gene and CRISPR/Cas9 RNP delivery [300]. LNPs are nanoparticles with a uni- or bi-phospholipid layer and an aqueous or lipophilic core for transporting drugs or nucleic acids. Although several difficulties in using LNPs as a delivery vehicle, such as prevention of degradation, avoiding being taken up by the mononuclear phagocyte system (MPS), reaching the target tissue, and escaping the endosome after intracellular uptake, have been encountered, they are still widely used for delivering mRNA vaccines [301]. Their advantages are that they are non-toxic and non-immunogenic and are easy to use, although the weak tissue tropism lowers their absorption by the target cells [302]. However, solutions to overcome this limitation are currently being sought. The microfluid mixing technique can be used to load diverse materials into LNPs by adjusting the core nanostructure [303], while selective organ targeting (SORT) offers a way to guide LNPs to the intended target, including the liver [304]. Furthermore, antibody coating has been utilized on LNPs targeting leukocytes [305]. Not only RNP structures but also loading plasmid DNA encoding Cas9 and gRNA, and Cas9 mRNA and gRNA separately, is possible [306]. While LNPs have not yet been applied to Wilson’s disease (WD), there are other examples of CRISPR-based gene editors for various liver diseases, such as Intellia Therapeutics’ NTLA-2001, which delivers mRNA-Cas9 to treat transthyretin amyloidosis caused by mutations in the TTR gene [307], and NTLA-2002 for the Hereditary Angioedema with the KLKB1 gene [308]. Hyaluronic acid-based nanoparticles, other polymeric nanoparticles, and gold nanoparticles have also been tested as transportation mechanisms in hepatocyte cell lines and hepatic disease models, showing adequate efficiency with low cytotoxicity [309,310]. As these transportation vehicles have been successfully applied in many gene mutation-related hepatic disorders, it seems likely that they will be applicable to WD treatment.

**Table 5 cells-13-01214-t005:** Recent gene-editing trials for treating WD.

Target Locus	CRISPR System	Associated Method	Delivery Method	Purpose	Model	Reference
c.2333G>T (p.R778L)	SpCas9	HDR with ssODN	Lipofection	Inducement of mutagenesis	Human embryonic stem cells (hESCs) and human-induced pluripotent stem cells (hiPSCs) in vitro	[199]
c.2333G>T (p.R778L)	SpCas9	HDR with ssODN	Electroporation	Repair	Human-induced pluripotent stem cells (hiPSCs) in vitro, ARG mouse ex vivo	[202]
c.1184delC (p.E396KfsX11)	SpCas9	HDR with ssODN	Lipofection	Inducement of mutagenesis, repair	HEK293T cells in vitro	[203]
Exon 8	SpCas9	HDR with ssODN	Lentivirus: SpCas9 AAV: sgRNA and donor template	Exon8 replacement	Mouse hepatocytes in vivo	[201]
c.2333G>T (p.R778L)	SpCas9, eSpCas9(1.1), SpCas9-HF1, and HypaCas9	HDR with ssODN	Lipofection	Inducement of mutagenesis	HEK293T and HeLa cells in vitro	[198]
c.2333G>T (p.R778L)	SpCas9 (WT and nCas), TALEN	HDR with ssODN	Lipofection	Inducement of mutagenesis	HEK293T and HeLa cells in vitro	[200]
c.3097G>A (p.T1033A) c.3659C>T (p.T1220M)	xBE3 xABE	–	Lipofection	Repair	HEK293T cells in vitro	[219]
c.1288dup (p.S430fs)	PE3	-	Electroporation	Repair	An organoid grown from a WD patient’s liver cells in vitro	[226]
Deletion of 900 bp of the coding region at the 3′ end and ~400 bp of the downstream untranslated region of *ATP7B*	–	–	Lentivirus	Transgene expression	Lymphatic endothelial cell (LEC) rat in vivo and hepatocytes ex vivo	[291]
Exon2	–	mini*ATP7B*	AAV8 and AAVAnc80	Transgene expression	*Atp7b*^−/−^ mice hepatocytes in vivo	[281]
Exon2, Exon8 (c.2248G>A)	–	mini*ATP7B*	AAV8	Transgene expression	*Atp7b*^−/−^ and *Atp7b*^tx-J^ mice in vivo	[283]
Exon2	–	Split-intein technology	AAV2/8	Transgene expression	HepG2 *ATP7B*-KO cells in vitro and *Atp7b^−/−^* mice in vivo	[284]
MBD 1-6	–	mini*ATP7B*	Lithium acetate	Eliminating each MBD to assess their respective roles	Yeast *Saccharomyces cerevisiae* in vitro	[282]

## 7. Conclusions

WD is a disease that can be managed well with good long-term survival and quality of life if diagnosed early. However, the limitations of cheap but inaccurate biochemical markers and accurate genotyping with exorbitant costs do not complement each other, which hinders early diagnosis. Furthermore, the current treatment guidelines necessitate lifelong medication therapy, supported by recommended diet restrictions, which may lead to low patient compliance that occasionally results in deteriorating outcomes. In this review, we have explored CRISPR/Cas system-based diagnostic methods for achieving accurate and economical diagnosis and discussed the potential use of CRISPR/Cas-related novel gene therapeutics combined with reliable delivery vehicles as alternative treatments, while also addressing some challenging issues that still need to be resolved.

Since the CRISPR/Cas system typically achieves specificity through the one-to-one correspondence between the gRNA and the target sequence, the need to design hundreds of gRNAs for targeting the numerous mutational variants in WD hinders the feasibility of CRISPR diagnostics. Another limitation is that there are no studies reporting the direct application of the CRISPR/Cas system to the diagnosis of WD patients. However, the CRISPR/Cas system has demonstrated its potential as a future genetic diagnostic tool in various human cell lines and animal models of WD. CRISPR/Cas system-based diagnostics offer several significant advantages over conventional diagnostic methods, including cost-effectiveness, time-saving, and user-friendliness. It is hoped that CRISPR/Cas system-based diagnostics will enhance the speed and accuracy of detecting genetic mutations. Techniques such as RGEN-RFLP, SHERLOCK, and DETECTR, which can yield results in under two hours, will significantly shorten the time needed for diagnosis compared to conventional methods like Sanger sequencing and NGS. Furthermore, the adaptability of CRISPR/Cas technology for use with minimal laboratory equipment could facilitate the expansion of point-of-care testing. This could be particularly transformative in remote or underserved areas, potentially improving early detection and treatment outcomes. Finally, CRISPR/Cas diagnostics can be easily customized to target specific genetic sequences by modifying the gRNA, thereby facilitating the development of tailored diagnostic tests for a diverse range of diseases. It is possible that future advancements may include multiplexed CRISPR diagnostics that can simultaneously test for multiple conditions, providing comprehensive health assessments in a single assay.

Despite the promising advances in CRISPR/Cas-based gene therapy, several limitations still need to be addressed. These include concerns about off-target effects, on-target cytotoxicity, the mandatory need for PAM sequence, the narrow editing windows, and target-specific delivery issues. Efforts to overcome these limitations involve engineering improvements in Cas proteins and sgRNAs to minimize side effects. Simultaneously, advancements in delivery systems need to enhance the targeting of the CRISPR/Cas system to specific tissues or cells while minimizing immunogenicity. It’s also essential to accurately assess the duration of activity of delivered editing systems in vivo to optimize gene editing efficiency while minimizing the risk of unintended small and large deletions, foreign gene insertions, or off-target effects. Therefore, comprehensive studies and evaluations across various target genes and patient populations are essential to ensure safety before CRISPR/Cas can be clinically applied with confidence.

## Figures and Tables

**Figure 1 cells-13-01214-f001:**
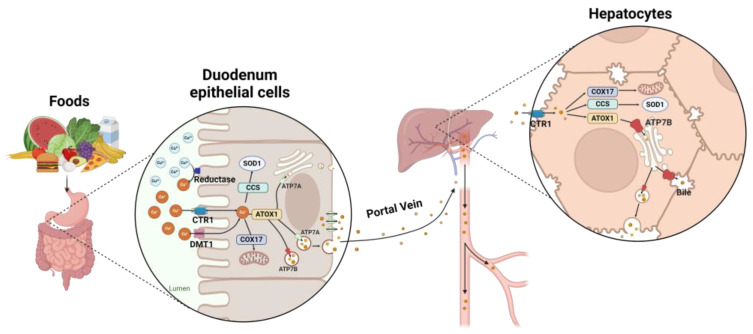
Copper uptake and homeostasis pathway in the human body. Copper is absorbed mainly in the duodenum and partly in the stomach via proteins such as high-affinity copper uptake protein 1 (CTR1) and divalent metal-ion transporter 1 (DMT1). Before transportation, cupric ions (Cu^2+^) are reduced to cuprous ions (Cu^+^) with the assistance of six transmembrane epithelial antigen of the prostate (STEAP) reductase, ascorbate, or duodenal cytochrome b (DCYTB). Upon absorption by and entry into an enterocyte, copper is transported by key copper chaperones, predominantly cytochrome c oxidase copper chaperone (COX17), superoxide dismutase (CCS), and antioxidant-1 (ATOX1). COX17 transfers copper to mitochondrial proteins SCO1 and SCO2, which are crucial for cytochrome c oxidase biogenesis. CCS delivers copper to superoxide dismutase (SOD1), which is a major antioxidant maintaining cytoplasmic ROS homeostasis. Copper is transferred by ATOX1 to the trans-Golgi network (TGN), where it interacts with copper-transporting P-type ATPase 1 (ATP7A) and P-type ATPase 2 (ATP7B). The ATP7A facilitates the delivery of Cu^+^ to copper-dependent enzymes such as peptidyl-α-monooxygenase, tyrosinase, and lysyl oxidase, while also exporting excess copper into the extracellular fluid. In contrast, the ATP7B stores Cu^+^ in the vesicles. Once exported, Cu^+^ is oxidized to Cu^2+^ by utilizing dissolved oxygen in the blood, where it binds to proteins such as albumin, histidine, etc., and then is transported via the portal vein to the liver. Hepatic uptake is facilitated by CTR1 post-reduction by copper reductase. Copper is transported within hepatocytes by key copper chaperones such as COX17, CCS, and ATOX1, likely entering via enterocytes. However, hepatocytes exhibit a high expression level of ATP7B and a low expression level of ATP7A. Consequently, ATP7B serves as the primary transporter in hepatocytes. The illustration was created using BioRender.com.

**Figure 2 cells-13-01214-f002:**
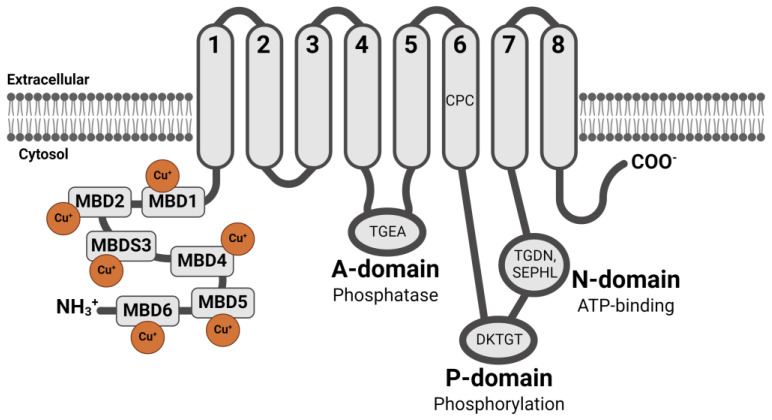
The structure of ATP7B. The A-domain has a TGEA motif, the P-domain has a DKTGT motif, and the N-domain has a TGDN and a SEPHL motif. There are six metal-binding domains (MBDs) containing the core conserved sequence GMXCXXC at the N-terminus located intracellularly, each of which can bind to one Cu^+^ ion. Among the transmembrane domains (TMDs), TMD6 has a CPC motif, which makes it characteristic of heavy metal-transporting P-type ATPase. The illustration was created using BioRender.com.

**Figure 3 cells-13-01214-f003:**
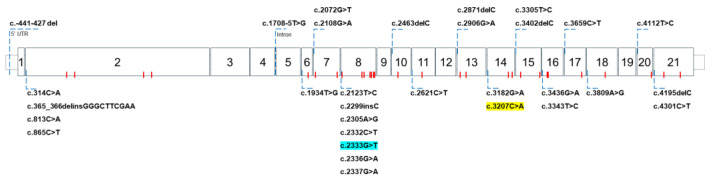
Prevalent mutation variants on the *ATP7B* gene that induce Wilson’s disease (WD). The most common mutations in the *ATP7B* genomic DNA causing WD are highlighted in blue (p.R778L/c.2333G>T; Asia) and yellow (p.H1069Q/c.3207C>A; Europe and North America). No predominant mutation was found to be associated with South America [28]. The p.H1069Q mutation occurs in the SEHPL motif in the N-domain, while the p.R778L mutation affects transmembrane transportation activity. The red dots indicate the locations of the mutation variants shown in Table 1.

**Figure 4 cells-13-01214-f004:**
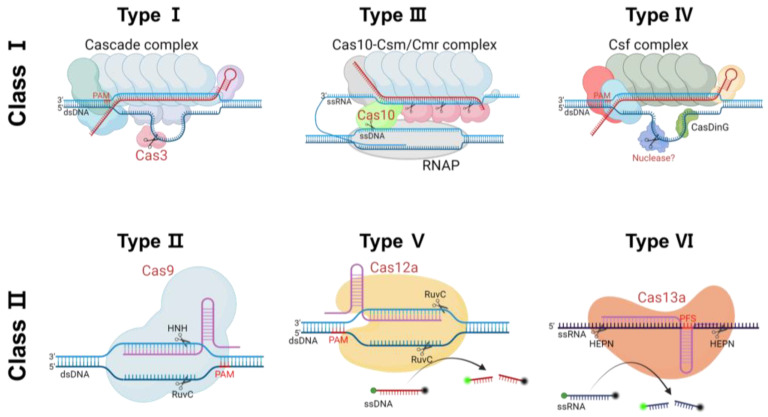
Schematics of the mechanisms of the CRISPR/Cas system types. The CRISPR/Cas system is classified into two classes based on the effector molecules involved. Class I is characterized by multi-unit effectors, while Class II exhibits a single effector. Each class is further divided into three types distinguished by their catalytic domain and target nucleotide specificity. Type I, represented by Cas3, forms a cascade complex with Cas6, Cas7, Cas5, Cas11, and Cas8. Upon crRNA binding, Cas3 generates a single-strand nick on the unwinding target DNA. Utilizing the Cas10-Cmr/Cms complex, the Type III system degrades nascent RNA before enzymatically cleaving complementary DNA. Type IV comprises csf1, csf2, csf3, an endoribonuclease (csf5), and a helicase (dinG), although its mechanism is not completely understood. The most widely used Type II system employs a single effector, Cas9, which induces a double-strand break (DSB) to target the double-stranded DNA (dsDNA) alongside the guide RNA (crRNA and tracrRNA). Types V and VI possess both target and collateral cleavage activities. Cas12 functions in conjunction with crRNA to generate staggered dsDNA breaks and nonspecifically cleaves the single-stranded DNA (ssDNA) collateral present in the vicinity of a dsDNA target. Cas13, in conjunction with crRNA, frequently cleaves single-stranded RNA (ssRNA) at multiple sites and exhibits nonspecific collateral ssRNA cleavage in the presence of the ssRNA target. The illustration was created using BioRender.com.

**Figure 5 cells-13-01214-f005:**
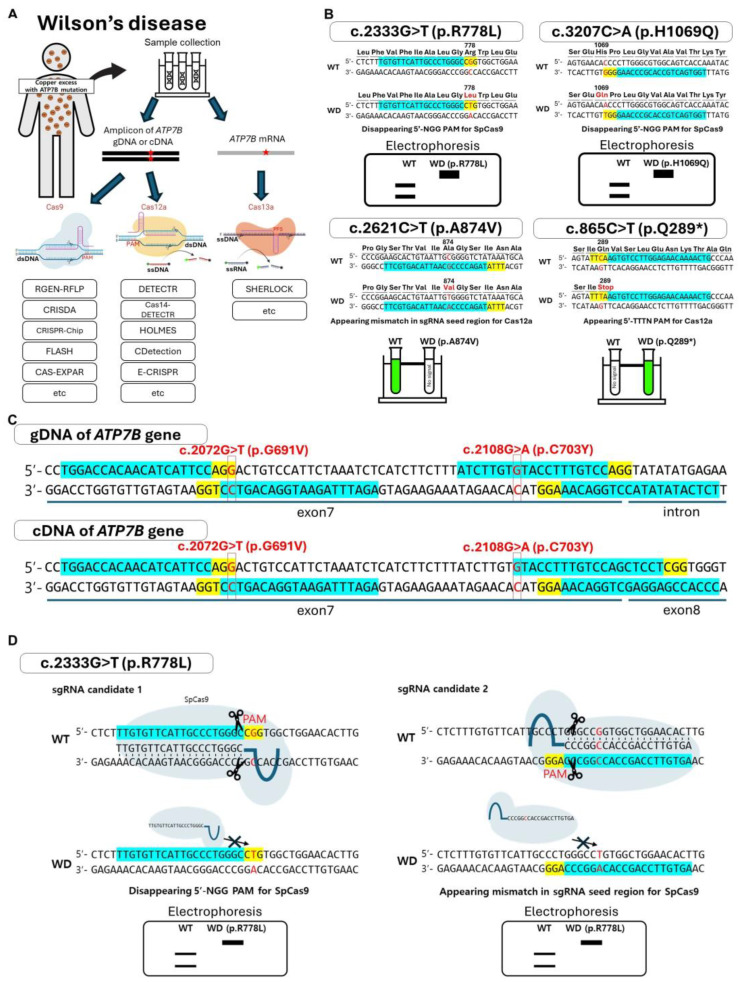
CRISPR/Cas system-based diagnosis of *ATP7B* mutations. (**A**) Diagnosing *ATP7B* mutations in WD patients depends on the specific type of mutation in the *ATP7B* gene and the type of nucleic acid (genomic DNA (gDNA) or mRNA) extracted from the patient. The red star represents the mutation point. The appropriate Cas protein must be selected and applied with these factors in mind. (**B**) A simple example of how the CRISPR/Cas system could potentially be used for diagnosing four distinct *ATP7B* mutations. The yellow highlight represents PAM sequences for SpCas9 (5′-NGG) and AsCas12a (5′-TTTN). The blue highlight represents the crRNA sequence in sgRNA to recognize the target sequence. The asterisk on the amino acid mutation means the stop codon. (**C**) The figure shows the locations in the genomic DNA (gDNA) and cDNA where the c.2072G>T (p.G691V) and c.2108G>A (p.C703Y) mutations can occur. The yellow highlight represents PAM sequences for SpCas9. The blue highlight represents potential sgRNA candidates for SpCas9, which are in proximity to the p.G691V and p.C703Y mutations. (**D**) The RGEN-RFLP method distinguishes between WT and mutant sequences by cutting the WT using sgRNA that matches the WT sequence. The upper figure shows possible scenarios after the treatment of SpCas9 with sgRNA. The blue highlight represents potential sgRNA candidates for SpCas9. The yellow marker indicates the location of the PAM sequence. The red letter on the DNA sequence represents the mutation site that induces p.R778L of *ATP7B*. The figure below shows the expected results of the RGEN-RFLP method resolved by agarose gel electrophoresis.

**Figure 6 cells-13-01214-f006:**
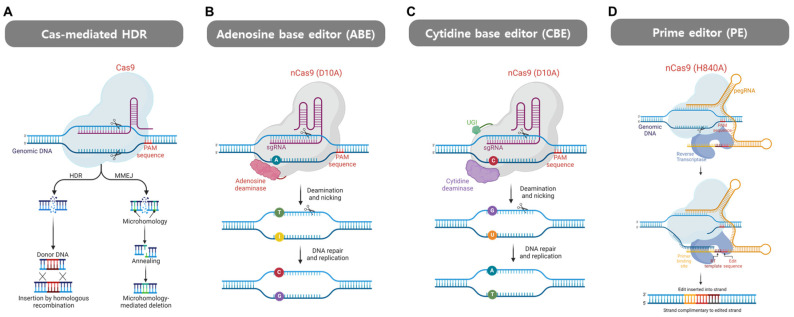
Schematics of the mechanism of four different CRISPR/Cas system-based treatment strategies. (**A**) To induce a homology-directed repair (HDR) event mediated by the Cas9 protein, it is necessary to provide donor DNA, which should include the wild-type sequence of the *ATP7B* gene. The Cas9 nuclease then induces a DSB in the vicinity of the mutation on the *ATP7B* gene. It is then necessary to select a suitable repair pathway for the treated cell. (**B**) Adenosine deaminase within the adenine base editor (ABE) system catalyzes the conversion of adenine (A) to inosine (I) on the strand not bound by the sgRNA. The nCas9 enzyme induces a nick in the DNA, initiating the DNA repair mechanism. Subsequently, during replication of the target DNA, a conversion of a thymine (T)/A base pair to a cytosine (C)/guanine (G) base pair is accomplished. (**C**) The cytosine base editing (CBE) is similar to that of ABE. In this case, a G/C-to-A/T base pair conversion is achieved via a uracil (U)-containing intermediate. Uracil glycosylase inhibitor (UGI) provides stability to the U/G base pair. (**D**) Unlike base editing, the prime-editing system utilizes a specific nCas9 variant with the H840A mutation, which silences the HNH domain. Reverse transcriptase (RT) then synthesizes a new DNA sequence from the end of the truncated strand using prime-editing guide RNA (pegRNA) containing both the template sequence and editing sequence. The illustration was created using BioRender.com.

**Table 1 cells-13-01214-t001:** Most prevalent *ATP7B* mutation variants by country.

Continent	Country [Reference]	Mutation Variant ^1^	Allele Frequency (%)	Exon	Nucleotide Change	dbSNP ^2^
Asia	China [33,34]	p.R778L	34.6–55.9	8	c.2333G>T	rs28942074
	Eastern China [35]	p.R778L	50	8	c.2333G>T	rs28942074
	Southern China [36,37]	p.R778L p.I1148T	18.93–23.29 8.74	8 16	c.2333G>T c.3443T>C	rs28942074 rs60431989
	India [38,39,40,41]	p.C271* p.R778W p.G1061E p.I1102T	10–19 16 12.1 12	2 8 14 19	c.813C>A c.2332C>T c.3182G>A c.3305T>C	rs572147914 rs137853284 rs764131178 rs560952220
	Southern India [42]	p.C271* p.G1061E	12 16	2 14	c.813C>A c.3182G>A	rs572147914 rs764131178
	Western India [43,44]	p.C271* p.E122fs	20.2 10.6	2 2	c.813C>A c.365_366 delinsGGGCTTCGAA ^3^	rs572147914 -
	Iran [45]	p.H1069Q	19	14	c.3207C>A	rs76151636
	Japan [46,47]	- p.R778L p.N958fs	11.0 13.4–20 15.9–20	Intron 4 8 13	c.1708-5T>G c.2333G>T c.2871delC	rs770829226 rs28942074 rs1957668488
	South Korea [48,49,50,51,52,53]	p.R778L p.A874V p.N1270S	37.5–49.3 8.3–19.4 12.1–14.9	8 11 18	c.2333G>T c.2621C>T c.3809A>G	rs28942074 rs121907994 rs121907990
	Lebanon [54]	p.M769fs	77.8	8	c.2304dup	rs137853287
	Saudi Arabia [55]	p.Q1399R	32	21	c.4195delC	rs886041336
	Taiwan [56]	p.R778L	43.1	8	c.2333G>T	rs28942074
	Vietnam [57]	p.S105* p.L1371P	32.27 9.09	2 20	c.314C>A c.4112T>C	rs753236073 rs1444841250
Europe	Austria [58]	p.H1069Q	36	14	c.3207C>A	rs76151636
	Bulgaria [59]	p.H1069Q	58.75	14	c.3207C>A	rs76151636
	Czech and Slovakia [60]	p.H1069Q	57	14	c.3207C>A	rs76151636
	Denmark [61]	p.W779X p.H1069Q	16 18	8 14	c.2336G>A or c.2337G>A c.3207C>A	rs137853282-3 rs76151636
	France [18,62]	p.H1069Q p.T1434M	14.8 24.32	14 21	c.3207C>A c.4301C>T	rs76151636 rs60986317
	East Germany [63]	p.H1069Q	63	14	c.3207C>A	rs76151636
	Greece [64]	p.R969Q p.H1069Q	12 35	13 14	c.2906G>A c.3207C>A	rs121907996 rs76151636
	Greece–Crete [65]	p.Q289*	88.8	2	c.865C>T	rs121907999
	Hungary [66,67]	p.H1069Q	46.8–64.3	14	c.3207C>A	rs76151636
	Continental Italy [68]	p.H1069Q	17.5	14	c.3207C>A	rs76151636
	Italy–Sardinia [68,69]	- p.M822fs p.V1146M	60.5 8.5 7.9	5′ UTR 10 16	c.-441_-427del c.2463delC c.3436G>A	rs879255499 - rs1213481140
	Latvia [70]	p.H1069Q	52.5	14	c.3207C>A	rs76151636
	Netherland [71]	p.H1069Q	32.9	14	c.3207C>A	rs76151636
	Poland [72,73]	p.H1069Q	66–71.5	14	c.3207C>A	rs76151636
	Romanian [74]	p.H1069Q	38.1	14	c.3207C>A	rs76151636
	Russia [75]	p.H1069Q p.A1135fs	39 19	14 15	c.3207C>A c.3402delC	rs76151636 rs137853281
	Serbia [76]	p.H1069Q	38.4	14	c.3207C>A	rs76151636
	Spain [77]	p.M645R	27.5	6	c.1934T>G	rs121907998
	Spain–Canary Islands [78]	p.L708P	64.6	8	c.2123T>C	rs121908000
	Sweden [75]	p.W779X p.H1069Q	16 38	8 14	c.2336G>A or c.2337G>A c.3207C>A	rs137853282-3 rs76151636
	Turkey [68]	p.T1220M	10	17	c.3659C>T	rs193922107
	United Kingdom [29]	p.M769V p.H1069Q	6 19	8 14	c.2305A>G c.3207C>A	rs193922103 rs76151636
Africa	Egypt [79]	p.C703Y p.N1270S	6.2 7.8	7 18	c.2108G>A c.3809A>G	rs767218895 rs121907990
North America	North American [75]	p.H1069Q	38	14	c.3207C>A	rs76151636
	United States [80]	p.H1069Q	40.3	14	c.3207C>A	rs76151636
South America	Brazil [81,82,83]	p.L708P p.H1069Q p.A1135fs	16.7 37.1 30.8–31.7	8 14 15	c.2123T>C c.3207C>A c.3402delC	rs121908000 rs76151636 rs137853281
	Venezuela [84]	p.G691V p.H1069Q p.A1135fs	9.6 7.7 26.9	7 14 15	c.2072G>T c.3207C>A c.3402delC	rs1555291801 rs76151636 rs137853281

^1^ The asterisk in the mutation variant column represents the stop codon induced by the nonsense mutation. ^2^ The Single Nucleotide Polymorphism Database accession number. ^3^ It has also been reported as c.365_366delinsTTCGAAGC [43].

**Table 2 cells-13-01214-t002:** Classification of the CRISPR/Cas system.

Class	Type	Subtype	Cleavage Domain	Substrate	Typical PAM/PFS ^1^
1	I	A, B, C, D, E, F, U	HD domain of Cas3 or Cas3”	dsDNA	3′-nt
	III	A, B, C, D	HD domain of Cas10, different from type I	ssRNA	Without PAM
	IV	Variants 1 and 2	Functionally uncharacterized	dsDNA	3′-DWN
2	II	A, B, C	RuvC, HNH	dsDNA	5′-NGG
	V	A, B, C	RuvC, NUC	dsDNA	5′-TTTN
	VI	A, B, C, D	HEPN	ssRNA	A given single base or without PFS

^1^ PAM, protospacer adjacent motif; PFS, protospacer flanking sequence.

**Table 3 cells-13-01214-t003:** Current treatment strategies for WD.

Treatments	Mechanism of Action	Advantages	Disadvantages
D-penicillamine (DPA) [169,170,171]	A copper chelator inducing cupruria. Upregulation of metallothionein.	The drug of choice for WD. Effectively alleviates hepatic symptoms.	Slow mitigation of neurologic symptoms can cause initial deterioration.
Trientine [172,173,174,175]	A copper chelator that induces fecal secretion and cupruria.	Alternative for DPA-intolerant patients. Used both for reversing symptoms or as a maintenance drug after withdrawal of DPA. Milder adverse side effects with lower incidence than DPA, including headache, arthralgias, myalgias, and abdominal pain.	Humidity control is required to store Trientine dihydrochloride salt (TETA 2HCL). The possibility of neurological worsening due to the rapid mobilization of large amounts of copper.
Bis-choline tetrathiomolybdate [174,175,176,177,178]	A decoppering agent inhibiting gastrointestinal absorption. Forms a complex with plasma copper and prevents intracellular uptake.	Prompt symptom reversal and alleviation. Useful for initial treatment of patients with neurologic symptoms Received orphan designation in the US and EU as a potential therapy for WD.	Infrequent but serious major adverse events, such as bone marrow suppression, aminotransferase elevation, and neurologic deterioration. There are currently no efforts to make the product commercially available.
Zinc [179,180,181,182,183]	Upregulates metallothionein in gastrointestinal epithelial cells, resulting in inhibition of gastrointestinal absorption and increase through fecal secretion.	Low cost in comparison to other treatments. In pediatric WD patients, zinc shows more efficient therapeutical effects than DPA and trientine.	There is a possibility of liver deterioration. Negative copper balance may be caused by inhibition of resorption of copper secreted in saliva, gastric juice, and intestinal secretions Exhibits mostly mild adverse events, but gastric irritation may be common (33.3%)
Sodium dimercaptopropane sulfate (DMPS) and Dimercaptosuccinic acid (DMSA) [184,185,186,187,188]	Forms complexes with heavy metals. Eliminates renal copper.	Decent neurologic symptom reversal in combination with zinc therapy. Relatively safe.	Should be aware of renal and dermatologic deterioration. Amount and quality of studies are limited.
Liver transplantation [189,190,191]	Replaces decompensated hepatic tissue.	Emergency treatment for acute liver failure.	Low engraftment efficiency in chronic liver disease. Requires lifelong immunosuppression. Not permanently effective.

**Table 4 cells-13-01214-t004:** The advantages and limitations of available CRISPR-based approaches for WD.

The Available CRIPSR-Based Approaches	Advantages	Disadvantages	Current CRISPR-Based Trials for WD
Homology-directed repair (HDR) with Cas9	Simple components. Applicable to a wide variety of vectors. Applicable to a wide range of mutations, including insertions, deletions, and all substitutions.	The potential detrimental effects of DSB, such as undesired edits by NHEJ, off-target effects, and small and large deletions. Low delivery efficiency of donor DNA.	[198,199,200,201,202,203]
Base editing (BE)	Non-DBS-inducing. Fewer off-target effects. No template DNA or RNA required.	Applicable to a single base pair substation only. Possibility of bystander base editing within an editing window.	[219]
Prime editing (PE)	Non-DBS-inducing. Fewer off-target effects. Applicable to a wide range of mutations including insertions, deletions, and all substitutions.	Low editing efficiency due to stability of pegRNA. Delivery issues due to too large size of prime editor protein or cDNA.	[226]

## Data Availability

The data used in this article were sourced from the materials mentioned in the References Section.

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
