# Peer review of "Navigating the CRISPR/Cas Landscape for Enhanced Diagnosis and Treatment of Wilson’s Disease"

_cells, 2024, doi:10.3390/cells13141214_

Round 1

Reviewer 1 Report

Comments and Suggestions for Authors

This is a timely and potentially very useful review for the clinical Wilson’s disease community.  While the standard approaches to Wilson's disease diagnosis and treatments as well as the emerging gene therapy have been been reviewed, the CRISPR/Cas technology has not yet been on the radar. Therefore, the summary presented by the authors is likely to be met with significant interestl.  The information is up-to-date, and I have only minor requests:

1. In the abstract (lane 16 and later lane 39):

[Wilson disease] … “is primarily caused by single-nucleotide polymorphisms in the ATPase copper transporting beta (ATP7B) gene” – please use “mutations” instead of polymorphisms as the later do not alter the protein function significantly – they are just normal variations

2.     Figure 1. This is a nice illustration; however it is somewhat outdated and, as shown, it suggests that the Cu transport mechanisms in intestine (where the reductase is shown but Cu chaperones are not) is very different from the liver, there no reductase is shown and chaperones are included.  The updated discussion of Cu homeostasis in enterocytes and hepatocytes can be found in the recent review by Lutsenko – see PMID 34734631. Please revise the figure to be more consistent.

3.     The summary Tables that the authors included are excellent.  It would be tremendously helpful if in addition to the table listing current CRISPR-based trials, the authors summarized (in a Table) the advantages and limitations of available CRISPR-based approaches.  Such discussion is included in the text, but a Table would provide the reader who is not intimately familiar with the field with a useful guide to choosing methodologies 

4.     The main advantage of using CRISPR in WD diagnosis instead of Sanger sequencing needs to be explained more clearly.  Given many different mutations of ATP7B (and sometimes more than one mutation on either or both DNA strands) would not be Sanger sequencing more straightforward and efficient way to determine which nucleotide was affected? 

5.     Related. In the compound heterozygous cases, when a combination of different mutations causes Wilson disease -  would multiple/different Cas proteins have to be used?  Would not that be cumbersome/expensive? The authors should spell out clearly when using CRISPR technology in WD diagnosis is advantageous (for major mutations -H1069Q and R778L?) and point out  limitations of the approach. 

6.     It would be interesting to hear the authors thoughts on how current limitations of CRISPR-based diagnostics for Wilson disease can be overcome 

7.     Figure 5 (Lanes 169-178). Figure 5 is very helpful, but needs to be expanded. Does the method shown in Fig. 5 work always or only in specific situations. i.e  is the PAM sequence always altered? It seems that there could be situations when it is not. Based on the author’s description, RGEN-RFLP approach might be able to overcome the challenges when PAM is not altered. It would be helpful to clearly state limitations of the method shown in Fig. 5 and also illustrate RGEN-RFLP.

Author Response

Thank you very much for your comments. Please find the response in the attachment.

Reviewer 2 Report

Comments and Suggestions for Authors

The authors Choi, Cha , and Kim provide an interesting review  in which the use of CRISPR/Cas for the diagnosis and treatment of Wilson disease is clarified. In general, the work provides interesting insights into the benefits of this pioneering method in various areas of medicine and outlines the possibilities for this rare hereditary metabolic disease.

Although common mutations in the ATP7B gene are discussed, I unfortunately miss a detailed discussion of how to use this method in several hundred known pathogenic mutations to improve diagnostic accuracy. If the main idea of the diagnostic utility of CRISPR/Cas is to improve validity, identification of just he main mutations will not suffice in the competition with other tests like REC or radio copper test.

The high costs of genetic diagnostics in general are also discussed, but the review does not provide an outlook on how the use of CRISPR/Cas could affect this in the short and long term. 

I also feel that the presentation of current therapeutic approaches lacks a more critical discussion of limitations caused, for example, by contradictory study results or limited scientific evidence. Overall, the description of Wilson disease does not go into too much detail, so I would recommend dispensing with the announcement of a detailed description in the introduction. 

In addition, I have the following notes / comments: 

Summary: 

“The condition is associated with the accumulation of copper in the body, leading to irreversible damage to various organs, including the liver, nervous system, brain, kidneys, and eyes.”

- Nervous system and brain are redundant. 

“However, the heterogeneous nature and individual presentation of physical and neurological symptoms in WD patients pose a major challenge to accurate diagnosis.”

- The clinical heterogeneity can be a challenge to even consider WD as a differential diagnosis. However, the disease is currently diagnosed with additional tests. This means that the problem of making a definitive diagnosis is due to the suboptimal accuracy of routine parameters or the cost of more informative tests such as REC (not described in this review) or genetic testing. 

“In addition, patients must follow a low-copper diet and/or take copper-chelating drugs for life.”

- It is highly unlikely that a low-copper diet alone will be sufficient to prevent disease progression. However, current practice guidelines recommend treatment for all WD patients, including asymptomatic patients. 

Introduction

"On the other hand, excess copper accumulation can cause oxidative damage, liver toxicity, and neurological disorders [6,7]."

This statement – though  correct - combines pathophysiological mechanisms (oxidative stress), with broad description of organ failure. Acute poisoning primarily leads to gastrointestinal symptoms, and in more severe conditions to multi-organ failure and hemolytic anemia, comparable to symptoms associated with WD. So far, I have not understood the relevance to juxtapose copper deficiency and copper intoxication at this very point. I would suggest to completely omit the descriptions of symptoms associated with copper intoxication und deficiency. The former will may be  described while specifying the clinical picture of Wilson disease anywhay, the latter does not provide critical information needed in this distinct context. 

"Its toxicity potential, which contributes to oxidative damage through a Fenton-like reaction, is well known [8], and it has recently been elucidated that it also induces stress on the endoplasmic reticulum, thereby leading to apoptosis [9] and copper-induced cell death by targeting the TCA cycle [10]."

- This is very true but incomplete: There are also data available suggesting that copper-induced damage is not purely driven by oxidative stress: Impact on mitochondrial membrane potential leads to breakdown of the TCA cycle even before reactive oxygen species are produced. Same applies to modulation of nuclear receptors. 

"Mutations disrupting the expression of any part of the amino acid sequence of ATP7B will curtail its activity, resulting in excessive amounts of copper in the body [13]. "

- I am not sure if I understood the sentence correctly. From my point of view, the sentence suggests that any mutation leads to a pathological variant of ATP7B, which indeed is not the case. As such, I suggest this sentence to be reframed. 

"Molecular genetic testing is recommended to confirm WD, especially if the patient has 57 unexplained neuropsychiatric disorders but no K-F rings [21]."

- I am not sure if this statement can be derived from [21]. Genetic testing is recommended especially if biochemical testing does provide definitive results.

Figure 3.

- Please consider adding the associated SNPs to the description (p.R778L / c.2333G>T; Asia) and (p. H1069Q / c.3207C>A; North-America and Europe). In South-America, there is no predominant mutation associated with WD, H1069 is responsible for about 5% (doi: 10.1016/j.ymgmr.2023.101034, [41]). Therefore, I would recommend specifying the statement to North-America, where H1069Q can be found in 50%.

4. CRISPR/Cas System-Based Gene-Editing Therapy Strategies for WD 

“Currently used treatments are mainly focused on decreasing copper absorption and increasing copper excretion. As copper is absorbed through the diet, it is necessary to change the diet to one low in copper. The usual daily limit for copper ingestion is 1 mg, which can be adjusted according to each patient’s condition after medical consultation.”

- This is correct for acute treatment, but please keep in mind for maintenance, copper restriction is nowadays regarded less strictly. 

Table 3:

- The need of vitamin B6 in DPA is not mandatory anymore except in cases of very high dosage. Trientine can be used both for acute and maintenance purposes. The statement of side effect frequency and neurologic function may be correct, but please keep in mind that studies comparing DPA and trientine partially provided conflicting results. Some trientine products do not require specialized storing condition. ATM: ATM is very unstable. As such bis-choline ATM has been used in clinical studies, bot development was terminated. Zinc: Primary function is reduced uptake by increased fecal secretion. It may take months of treatment to achieve a a negative copper balance. Hence, while zinc may be an option to eliminate stored copper, the statement is misleading. Side effects are mild, but gastric side effects may occur in up to 30%. Since this medication is especially employed in asymptomatic patients, there is a relevant risk of reduced adherence even though side effects are mild, compared to chelating agents. DMPS: even though published Data on DMPS + Zinc are interesting, please keep in mind that – compared to DPA and trientine – amount and quality of available studies are lower. 

Author Response

(The authors gave the same response as above.)

Round 2

Reviewer 2 Report

Comments and Suggestions for Authors

I thank the authors for the adaptation of their manuscript. There are only a few minor points to be considered, which relate to Table 3:

1.  There are formulations of TETA 2HCL that no longer require refrigeration. 

2. I am not aware of any studies showing better neurological preservation with trientine than with bis-choline TTM. 

3. Bis-choline TTM is no longer being considered for the market. I am not sure if the lack of clinical trials is the main reason. Although the background does not need to be discussed in this paper, I would still recommend changing the sentence from "Still an investigational drug and more clinical trials are needed to make it commercially available" to something like "There are currently no efforts to make the product commercially available".

4. Zinc: There are publications on (particularly) liver deterioration following treatment with zinc. However, zinc is still considered an effective treatment. It may also be worth noting that trientine is much more expensive than penicillamine, which is more expensive than zinc. 

Author Response

(The authors gave the same response as above.)
